Medical Imaging with Deep Learning 1–4, 2023

# Improving Zero-Shot Detection of Low Prevalence Chest Pathologies using Domain Pre-trained Language Models

**Aakash Mishra**[*][1]                                    AMISHRA@COLLEGE.HARVARD.EDU

**Rajat Mittal**[*][1]                                       RAJATMITTAL@COLLEGE.HARVARD.EDU

**Christy Jestin**[1]                                      CHRISTYJESTIN@COLLEGE.HARVARD.EDU

**Kostas Tingos**[1]                                       KOSTASTINGOS@COLLEGE.HARVARD.EDU

**Pranav Rajpurkar**[1]                                    PRANAV_RAJPURKAR@HMS.HARVARD.EDU

[1] *Harvard University*

## Abstract

Recent advances in zero-shot learning have enabled the use of paired image-text data to replace structured labels, replacing the need for expert annotated datasets. Models such as CLIP-based CheXzero utilize these advancements in the domain of chest X-ray interpretation. We hypothesize that domain pre-trained models such as CXR-BERT, BlueBERT, and ClinicalBERT offer the potential to improve the performance of CLIP-like models with specific domain knowledge by replacing BERT weights at the cost of breaking the original model's alignment. We evaluate the performance of zero-shot classification models with domain-specific pre-training for detecting low-prevalence pathologies. Even though replacing the weights of the original CLIP-BERT degrades model performance on commonly found pathologies, we show that pre-trained text towers perform exceptionally better on low-prevalence diseases. This motivates future ensemble models with a combination of differently trained language models for maximal performance.

**Keywords:** Zero-Shot, Fine-Tuning, Pre-Training, Multi-Modal, Self-Supervised Learning

## 1. Introduction

Contrastive learning methods have helped create deep learning models with robust zero-shot performance in the field of medical imaging. Innovations in deep learning have enabled the automation of tasks such as medical image interpretation (Litjens et al., 2017; Rajpurkar et al., 2017). Many of these methods, however, rely on the existence of a large dataset of annotated examples. These annotations often take a significant amount of expert time to assign, and the resulting approaches are limited to only predicting diseases that are explicitly annotated (Smit et al., 2020). Recently, the emergence of contrastive learning approaches for multimodal zero-shot learning have allowed for paired image-text data to replace structured labels and still achieve competitive performance on many downstream tasks (Radford et al., 2021). Some models have even achieved expert-level, zero-shot chest X-ray pathology classification performance (Tiu et al., 2022).

While many models such as CheXzero are initialized with weights derived from natural image-caption training, there have been a number of open-source language models that specialize in specific medical domains such as CXR-BERT (Boecking et al., 2022), BlueBERT (Peng et al., 2019), and ClinicalBERT (Alsentzer et al., 2019). In particular, domain pre-trained models see a larger medical corpora than is available in chest X-ray reports, and

---

[*] Contributed equally

as a result, form richer embeddings for low-prevalence pathologies (infrequently mentioned pathologies in the image-report dataset). The trade-off is that using a pretrained text tower breaks the alignment between the vision and language modalities. We hypothesize that these domain pre-trained models, having seen a larger medical corpus, offer the potential to improve the performance of CLIP-like models on low-prevalence pathologies. In this work, we evaluate the performance of zero-shot classification models with domain-specific pretraining and find that they perform especially well for detecting low-prevalence pathologies.

## 2. Methods

We employ contrastive learning with image–text pairs to achieve zero-shot multi-label classification, utilizing two embedding models: a vision embedder and a text embedder. The embedding models are trained via a contrastive loss and the similarity is assessed. Instead of initializing both towers with CLIP weights like the baseline model, CheXzero (Tiu et al., 2022), we keep the existing vision tower from the generalized pretrained model and the replace the CLIP text tower with a domain-specific language model. We employ three language domain pretraining models: CXR-BERT, BlueBERT, and ClinicalBERT, all of which were pretrained on PubMed and MIMIC-III (Peng et al., 2019; Alsentzer et al., 2019; Boecking et al., 2022). CXR-BERT was further pretrained on MIMIC-CXR to specialize in the chest X-ray domain. Our text stack includes a text projection to complete our pretrained models and produce a final embedding size of 128 since originally the embedding size for each of the three language models was not standardized.

We then trained each model on the contrastive alignment task for 5 epochs using MIMIC-CXR data and then proceeded with a zero-shot evaluation on the chest X-ray pathology classification task, using negative (e.g. no pneumonia) and positive prompts (e.g. pneumonia) to calculate similarities between the image and prompts as logits for probability of occurrence with the VinDr-CXR dataset. We follow the same zero shot inference strategy as Tiu et al. (2022). Furthermore, we trained a single CheXzero baseline model from CLIP weights to provide a better comparison for our single-model experiments since the published model is an ensemble. All models were trained for 5 epochs with a batch size of 32, a learning rate of 5e-6, and a SGD optimizer with a momentum of 0.9.

## 3. Results

Both CLIP-vision + domain pre-trained language models (ClinicalBERT and CXRBERT) exceed the AUC performance of the baseline in every category of low-prevalence pathology from the VinDr-CXR dataset which contains labels for many rarely mentioned diseases, including lung tumor, aortic enlargement, enlarged pulmonary artery, and clavicle fracture, all of which occur as strings less than 100 times in the more than 300 thousand report impressions in MIMIC-CXR. The opposite trend holds for high-prevalence diseases where the baseline outperforms the majority of mixed CLIP-vision + domain pretrained text models. As reflected in Table 1, we see that for rare diseases where the number of occurrences is below 1% and 100 total mentions, the best model is the CXR-BERT hybrid with an improvement of 0.15 over the CheXzero baseline. The second-best model is ClinicalBERT with an AUC score improvement of 0.08 on average across the low prevalence disease mentions. Both ClinicalBERT and CXR-BERT outperform our baseline in each category. For high-prevalence

| Configuration | Lung tumor | Aortic enlargement | Enlarged PA | Clavicle fracture |
|---|---|---|---|---|
| **Occurrences** | 1 ($\leq 1\%$) | 6 ($\leq 1\%$) | 38 ( $\leq 1\%$) | 69 ($\leq 1\%$) |
| Baseline | 0.687 (0.633, 0.725) | 0.580 (0.547, 0.619) | 0.713 (0.557, 0.838) | 0.664 (0.401, 0.848) |
| BlueBert | 0.654 (0.605, 0.699) | 0.640 (0.609, 0.670) | 0.667 (0.537, 0.805) | 0.585 (0.264, 0.809) |
| ClinicalBert | 0.713 (0.665, 0.753) | 0.596 (0.567, 0.628) | 0.757 (0.580, 0.891) | 0.762 (0.445, 0.978) |
| CXRBert | **0.714 (0.666, 0.751)** | **0.760 (0.737, 0.782)** | **0.854 (0.777, 0.935)** | **0.853 (0.839, 0.872)** |

| Configuration | Pleural effusion | Pneumonia | Atelectasis | Pneumothorax |
|---|---|---|---|---|
| **Occurrences** | 78644 (18.6%) | 65204 (15.7%) | 56377 (13.6%) | 45994 (11.1%) |
| Baseline | 0.864 (0.823, 0.897) | **0.849 (0.823, 0.874)** | **0.770 (0.727, 0.814)** | **0.811 (0.762, 0.865)** |
| BlueBERT | 0.857 (0.821, 0.882) | 0.540 (0.506, 0.580) | 0.732 (0.680, 0.779) | 0.689 (0.591, 0.783) |
| ClinicalBERT | 0.773 (0.732, 0.805) | 0.821 (0.799, 0.845) | 0.639 (0.589, 0.699) | 0.590 (0.490, 0.687) |
| CXRBERT | **0.895 (0.871, 0.918)** | 0.819 (0.801, 0.841) | 0.675 (0.624, 0.714) | 0.683 (0.602, 0.782) |

Table 1: Pretrained text tower zero-shot bootstrapped AUC performance on low-prevalence pathologies in the MIMIC-CXR training set, evaluated on the VinDr-CXR dataset. Both ClinicalBERT and CXRBert outperform the baseline on the majority of low-prevalence pathologies. The same cannot be said for common pathologies. Occurrences are defined as the number of MIMIC-CXR reports with impressions that contain the name of the pathology. Format: AUC values and 95% confidence intervals

pathologies, we see that the CheXzero baseline tends to outperform the hybrid CXR-BERT AUC by an average of 0.05 and the ClinicalBERT AUC performance by 0.12.

## 4. Discussion

**Domain pretrained text towers have extremely strong performance on rarely mentioned diseases.** We find that pre-trained text towers perform extremely well across categories with little mention in the reports in the MIMIC-CXR alignment training dataset. This motivates the need for pre-trained text towers depending on the use-case of a zero-shot classifier. When diseases have little-to-no mention in the alignment training dataset, the baseline is automatically at a disadvantage. At zero-shot classification time, in the ideal scenario, the model will produce a relevant image embedding containing information about the new pathology and the text tower will embed the new pathology to an aligned encoding. However, if the text tower has rarely or never been exposed to the new pathology as is the case with CheXzero, which was initialized with CLIP weights, the likelihood of the text embedding being meaningful in reference to the image embedding is small. Meanwhile, pre-trained text towers that have been trained on large datasets like MIMIC III and PubMed abstracts have encountered impressions of these low-prevalence pathologies in their masked-language modeling pre-training task. The surprising result is that even after alignment training where the pathology is rarely mentioned, the text embedding model is still able to perform well on these rarer categories.

Our research suggests that a zero-shot classifier designed to classify a broad range of both common and low-prevalence chest X-ray pathologies will likely need some form of a domain pretrained text tower. One option is to include pretraining in the text encoder in a way that does not hurt performance on common diseases, a task of future research. The other option is a weighted ensemble that has both a domain pretrained language model and a general BERT model.

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
