# OpenReview forum: "Improving Zero-Shot Detection of Low Prevalence Chest Pathologies using Domain Pre-trained Language Models"
_MIDL.io/2023/Short_Paper_Track — MIDL 2023 Short paper track Poster_

### Official Review · Reviewer_yVyx · 2023-04-20
**Paper 120 review**

**Rating:** 6
**Confidence:** 4

**Review:**

The basic premise of the paper is good but it has not been explained properly.

---

### Official Review · Reviewer_my2K · 2023-04-24
**Effective use of pretrained language models**

**Rating:** 8
**Confidence:** 3

**Review:**

This paper evaluates different language models, pre-trained on general corpuses, for disease detection when image+text are available. The authors focus on rare disease, which are difficult to generalize to and have no training data.

While I am not quite sure that the term of zero-shot is right, clearly the general models are able to reach very impressive performances "out of the box", even better than the authors' domain specific model (CheXzero).